# The Replacement of Ground Corn with Sugar Beet in the Diet of Pasture-Fed Lactating Dairy Cows and Its Effect on Productive Performance and Rumen Metabolism

**DOI:** 10.3390/ani12151927

**Published:** 2022-07-28

**Authors:** Juan Pablo Keim, Jonattan Mora, Sebastián Ojeda, Bernardita Saldías, Ulrike Bedenk

**Affiliations:** 1Animal Production Institute, Faculty of Agricultural Sciences, Universidad Austral de Chile, Valdivia 5110566, Chile; jonattan.mora@alumnos.uach.cl; 2KWS Chile Ltda., Longitudinal 5 Sur. km 79, Rancagua 2820000, Chile; sebastian.ojeda@kws.com; 3Independent Researcher, Lincoln 7608, New Zealand; bernarditasaldias@gmail.com; 4KWS SAAT SE & Co. KGaA, Grimsehlstrasse 31, 37574 Einbeck, Germany; ulrike.bedenk@kws.com

**Keywords:** *Beta vulgaris* ssp. *vulgaris*, N excretions, pasture fed cows

## Abstract

**Simple Summary:**

Cereal grains have increased in cost, and therefore dairy farmers try to find alternatives to provide energy in the rumen. Sugar beet roots have high energy content and may be a sound alternative to replace ground corn partially or totally in the diet of dairy cows. Thus, the aim of this study was to evaluate the replacement of ground corn with fresh sugar beet on the milk production responses, rumen metabolism, and profitability of pasture fed dairy cows. Although cows supplemented with sugar beet roots had reduced dry matter intake and milk production compared with control cows, fat percentage was increased, and therefore, there were no differences in energy corrected milk yield (FPCM) among the treatments. Moreover, feeding costs were reduced, and thus, the margin over feed costs was increased for sugar beets. In addition, the replacement of ground corn by sugar beets reduced urinary N excretion, and thus, it may contribute to the reduction in N_2_O emission from dairy systems. Using sugar beet roots as an energy supplement can be a suitable alternative to ground corn in pasture-fed lactating dairy cows, increasing the sustainability of dairy systems.

**Abstract:**

(1) Background: Sugars have a potential to provide great amounts of fermentable energy in the rumen. Feeding fresh sugar beet (SB) to dairy cattle to replace a portion of the grain in the ration has not received sufficient attention. This study determined dry matter intake (DMI), feeding behavior, rumen fermentation and milk production responses when replacing corn grain with increasing levels of SB in pasture-fed lactating dairy cow diets. (2) Methods: A total of 12 early-lactation cows were used in a replicated (n = 4) 3 × 3 Latin square design. The control diet consisted of 21 kg dry matter (DM) composed of 6.3 kg DM green chopped perennial ryegrass, 7 kg DM grass silage, 2 kg DM of concentrate, 1 kg DM soybean meal and 4.5 kg DM of ground corn. The other treatments replaced 50% or 100% of the ground corn with SB roots. (3) Results: The replacement of ground corn with sugar beet reduced DMI and milk yield (*p* < 0.05), but it increased milk fat concentration (*p* = 0.045), reduced feeding costs and increased margin over feed costs (*p* < 0.01). Urinary nitrogen was linearly reduced with SB supplementation (*p* = 0.026). (4) Conclusions: Using SB roots as energetic supplement can be a suitable alternative to ground corn in pasture-fed lactating dairy cows.

## 1. Introduction

In grazing dairy systems in humid temperate regions, the supply of metabolizable energy is the first-limiting factor for milk production from high-quality pasture [1]. Thus, concentrate supplementation is used to increase total dry matter intake (DMI) and energy intake relative to that achieved with pasture-only diets [2]. There are several sources of energy supplementation, with cereal grains such as corn and wheat as the main energy source in diets of high-producing dairy cows because they are cost-effective sources of digestible energy [3]. The primary aim of feeding grains to ruminants is that their high carbohydrate content in the form of starch enables high energy density in their diets to support milk production [4] and to increase milk protein percentage [5]. However, cereal grains are used in other production systems such as pork and poultry, industries for human food production and biofuel production, which compete with ruminant systems, increase the demand for cereals and explains in part the price instability over time that has resulted in a 60% increase over the last four years [6]. 

Efforts have been made to find alternatives that increase the energy density of rations. Among those alternatives, sugars are water-soluble carbohydrates that are readily degradable in the rumen and have a potential to provide great amounts of fermentable energy that enhances microbial protein production [7]. As reviewed by Oba [7], the nutritional characteristics of sugars allows for the use of high-sugar feedstuffs as an alternative energy source for lactating dairy cows to increase dietary energy density with reduced risk of rumen acidosis. Rumen acidosis occurs mainly due to excess starch supplementation [8]. Propionate production from starch fermentation involves the conversion of pyruvate to propionate via lactate and acrylate, where lactate may be produced in appreciable amounts, reducing rumen pH [9]. Conversely, feeding high-sugar diets often increases DMI [10] and butyrate concentration in the rumen [11,12]. Butyrate is produced in the rumen by: (1) the ß-oxidation of fatty acids through the condensation of 2 mol acetate into 1 mol butyrate, a pathway that occurs in *Butyrivibrio fibrisolvens*; (2) the carboxylation of acetyl-CoA to malonyl-CoA and then reaction with acetyl-CoA to yield acetoacetyl-CoA [9]. In terms of rumen microbiome, it has been observed that under in vitro conditions, high sugar levels increase the relative abundance of *Bacteroidetes* and *Firmicutes* phyla, reducing *Proteobacteria*; in contrast, at the genus level, the relative abundance of *Treponema* increased and *Ruminobacter*, *Ruminococcus* and *Streptococcus* decreased [13].

Typical high sugar feedstuffs are fodder and sugar beets [14], swedes [15], high-sugar grasses [16]; whey products and molasses [7]. Feeding sugar beet (SB) roots to dairy cattle to replace a portion of the grain in the ration is a concept that has not received sufficient attention [17]. Sugar beets *(Beta vulgaris* ssp. *vulgaris)* are characterized by both high yields per hectare and high sucrose content. Although primarily grown for sugar production, SB and their by-products can also be used for feeding ruminants [18]. Sugar beet roots can be offered fresh or ensiled and contain 236 g dry matter (DM)/kg, 760 g sugar/kg DM, 61 g crude protein (CP)/kg DM and 125 g neutral detergent fiber (NDF)/kg DM [17]. Due to their high DM yields (>20,000 kg DM/ha for the roots), sugar beets provide a considerable advantage in terms of cost of production per kg DM when compared with other energy-rich feedstuffs [17,19].

There are studies that have evaluated the inclusion of fresh or ensiled sugar beets [20,21,22,23] and fodder beets [24,25,26,27] in dairy cow diets. Few studies have involved the feeding of high concentrations of sugar from beets to replace starch in the diet. Evans et al. [22] provided dairy cows with diets that contained 0, 80, 160 or 240 g/kg of the total ration DM as fresh, chopped sugar beets and reported no losses in milk production, milk composition or DMI when compared with the control ration in which the concentrate was based on corn and barley. Evans and Messerschmidt [17] suggest that fresh beets might not pose a greater risk of digestive or metabolic upset than grains when presented as a portion of a total mixed ration. Same authors indicated that more studies are required to assess the effects of providing beets in early lactation dairy cows. Moreover, to the best of our knowledge there is less evidence of replacing starch by sugar in grazing early lactating dairy cows. Thus, the aim of this study was to determine DMI, feeding behavior, rumen fermentation, milk production responses and nitrogen (N) partitioning when replacing ground corn with increasing levels of sugar beet in pasture-fed lactating dairy cow diets. We hypothesized that the partial or total replacement of corn grain with sugar beet in early-lactation pasture-fed dairy cows would not negatively affect DMI, rumen fermentation, milk production or its composition. However, it will increase economic response.

## 2. Materials and Methods

The study was conducted at the Experimental Research Station of Universidad Austral de Chile, Valdivia, Chile, between June and August 2020. All experimental procedures were approved by the Universidad Austral Institutional Animal Care and Use Committee (Approval Number: 393/2020).

### 2.1. Animals, Housing and Experimental Design

Cows were subjected to a fifteen-day uniformity period where all cows were offered the control diet. The animals were selected according to milk production in the previous lactation and by milk production and body weight (BW) prior to the experiment (30.6 kg milk/d, 592 kg BW and 73 days in milk). Twelve multiparous lactating dairy cows were randomly allocated to the three dietary treatments according to milk production measured during the uniformity period; thus each experimental group covered the whole range of milk production. The experimental design was a replicated (n = 4) 3 × 3 Latin Square design with three 21 d periods. Each experimental period consisted of 14 d of adaptation to diets and 7 d of experimental measurements. Cows were grouped in four squares according to milk production and BW. The sequencies were defined according to [28] and balanced for residual effects of previous treatments; thus, all combinations for treatments (100C to 50C-50SB, 100C to 100SB, 50C-50SB to 100C, 50C-50SB to 100SB, 100SB to 100C and 100SB to 50C-50SB) occurred when changing cows from a previous experimental period to the other.

All animals were housed in the same tie-stalls barn, equipped with rubber bedding and individual feeders. Animals were milked twice daily at 0700 and 1600 h in the milking parlor close to the barn facility; the animals had ad libitum access to water. The control group (100-C) received a diet similar in type and quantity of feeds offered to cows in commercial dairy farms from southern Chile, aiming to supply their energy and protein requirements (21 kg DM composed of 6.3 kg DM green chopped perennial ryegrass, 7 kg DM grass silage, 2 kg DM of formulated concentrate without corn, 1 kg DM soybean meal and 4.5 kg DM of ground corn). The same feeds were offered to the two treatment groups, but in addition 2.5 kg DM (50C-50SB: replacement of 50% of ground corn by sugar beet) or 5 kg DM (100-SB: replacement of 100% of ground corn by sugar beet) of ground corn were replaced by sugar beet. An additional 0.25 and 0.5 kg DM of SB were added for 50C-50SB and 100SB compared with the amount of corn offered, to keep diets isoenergetic. The formulated concentrate was based on small grain cereals and by-products, but no corn grain was included. A 250 g mineral mixture was offered along with the diets. Prior to the beginning of the experiment, all feed ingredients were analyzed for their chemical composition (Table 1), and thereafter the nutrient concentration of the commercial concentrate was adjusted for each treatment to keep diets isoenergetic and isonitrogenous with a daily supply of 35.5 Mcal NE_l_ and 3.5 kg CP.

Prior to feeding, all feeds were weighed and mixed according to each cow’s dietary treatment and offered after a.m. and p.m. milkings. Every 15 days, sugar beet (variety Glacita KWS) roots were harvested on a commercial farm and transported to the research station. Sugar beet roots were manually chopped with shovels, aiming for a particle size of 5–10 cm. Perennial ryegrass forage (3–3.5 leaves/tiller stage of growth) was harvested with a chopper machine 10 cm above ground level and offered fresh.

### 2.2. Intake and Ingestive Behavior

Feeds offered and orts were recorded all days of week 3 of each period. Sub-samples off each feed ingredient and orts were collected and their DM content determined in a forced-air oven at 105 °C for 12 h. The nutrient intake of each feed ingredient was calculated by multiplying the DMI of each ingredient and its nutrient concentration; then, that was used to calculate total nutrient intake. Samples of grass silage, perennial ryegrass forage and sugar beet roots were collected once a week, freeze-dried, ground through a 1 mm screen (Wiley Mill, Philadelphia, PA, USA) and stored for chemical analyses. Samples for the chemical analyses of commercial concentrate and soybean meal were taken once per experimental period. For each sample, ash and lipids were analyzed according to [29] (methods 942.05 and 920.39 for ash and EE, respectively); N content was determined by combustion (Leco Model FP-428 Nitrogen Determinator, Leco Corporation, St Joseph, MI, USA) and was used to calculate CP content (N × 6.25). Neutral detergent fiber was determined as aNDFom [30] using sodium sulfite (Merck) heat-stable amylase (Ankom Technology Corp., Macedon, NY, USA) and expressed exclusive of residual ash and ADFom according to [29] (method 973.18) expressed exclusive of residual ash. 

Behavior was evaluated based on the following activities: eating (bite procurement, chewing between bites, and/or searching), rumination (standing or lying behavior of the animal) and other activities (resting, social interactions, drinking, demonstration of estrus, and others). Each activity was visually and continuously recorded by trained operators on d 4 of each experimental week, each operator working in sequences of 6 h. Every 10 min, from 0800 h to 1600 h and 1700 h to 0700 h (presence of the cows in the barn), they noted the activity that each cow was performing. Each activity’s time was calculated by the summation of all 10-min intervals of activity.

### 2.3. Milk Production, Composition and Body Weight

Cows were milked at 0700 h and 1600 h, and milk yield was recorded daily with a flow sensor (MPC580 DeLaval, Tumba, Sweden) during the experimental periods. The daily average for the final week of each period is reported. Only the data from the sampling periods were used in the statistical analysis. Body weight was measured every day after each milking with an automatic weigh scale (AWS100 DeLaval, Tumba, Sweden) while cows were returning to the barn. To determine BW change, BW was averaged for the first and last three days of each experimental period. Body condition score (BCS) was registered at the beginning of the experiment and the last day of each experimental period, always by the same operator based on a 5-point scale [31].

Milk samples (350 mL) were collected with milk meters (Waikato MK V, Waikato, New Zealand) at morning and afternoon milking times on d 15, 18 and 21 of each experimental period for fat, protein and milk urea analyses by mid-infrared spectrophotometry (Foss 4300 Milko-scan, Foss Electric, Hillerød, Denmark). Fat–protein corrected milk was calculated according to [32].

### 2.4. Rumen Fermentation

Rumen fluid was collected by inserting a stomach tube (Flora Rumen Scoop; Prof-Products, Guelph, ON, Canada) into the rumen at 0800 h, 1200 h and 1700 h on d 20 of each experimental period.

To reduce saliva contamination, the first portion of the liquor collected was discarded. Rumen liquor was strained through four layers of cheesecloth. A 10-mL sample was drawn off, mixed with 0.2 mL of 50% (*w*/*v*) sulphuric acid and stored at −20 °C for the further determination of volatile fatty acid (VFA) and ammonia (NH_3_) concentrations. Rumen fluid was allowed to thaw for 16 h at 4 °C and then centrifuged at 10,000× *g* for 10 min at 4 °C. An amount of 6 mL of supernatant was drawn off and then centrifuged at 10,000× *g* for 10 min at 4 °C. Thawed supernatant of rumen fluid samples was analyzed for VFA using gas chromatography as described by [33] (Shimadzu GC-2010 Plus High-end GC, equipped with GC capillary column, SGE, BP21 (FFAP), temperature range = 35 °C to 240/250 °C, UOM = EA), and for NH_3_ by the phenol-hypochlorite reaction method [34]. Total VFA (tVFA) was considered the sum of acetate, butyrate, propionate, isobutyrate, valerate, isovalerate and caproate. Minor VFA was considered the sum of isobutyrate, isovalerate, valerate and caproate.

For the whole experiment, rumen pH was monitored using a wireless telemetric pH bolus (eCow, Exeter, UK), that were previously validated. The pH boluses were calibrated before use, programmed to measure rumen pH at 15-min intervals and inserted directly into the ventral sac of the rumen of each cow. The pH data were reported as mean pH, pH per hour and times with pH > 6.2, between 5.8–6.2 and pH < 5.8.

### 2.5. Urine Collection Microbial Protein Synthesis and N Balance

Rumen microbial N flow was estimated based on purine derivatives (PD). Spot urine samples (20 mL) were collected by subvulvar stimulation every 3 h during d 19 in each experimental period. Samples were acidified with 2 mL sulphuric acid (10% *v*/*v*) and stored at −20 °C. A composite sample per cow was made for each period and analyzed for allantoin, uric acid and creatinine by HPLC. A Waters Alliance 2996 sensitive module HPLC (Waters, Milford, MA, USA) equipped with a UV spectrophotometric detector set at 220 nm was used for these analyses. The stock standard solutions (1 mg/mL) of allantoin and creatinine were freshly prepared in water. The uric acid standard was dissolved in water (1 mg/ML) by adding 0.01 N sodium hydroxide solution (5 mL/100 mL stock standard solution) to make the pH 7. The quantitative HPLC separations were performed at a temperature of 30 °C on a C18 reversed-phase column (250 × 4.60 mm^2^ I.D., 5 µm particle size). The mobile phase was 10 mM potassium dihydrogen phosphate solution (pH 7.0). The flow rate was 1 mL/min, and the absorbance detection was set at 220 nm. Compound peaks were identified by the retention times and quantified by comparison of the peak areas of the samples with those of authentic standards on a 20 mL injection.

The equations used in calculating the estimated microbial N supply outlined below have been described previously [35,36]. 

The PD index was calculated based on total PD [allantoin (mmol/L) + uric acid (mmol/L)] as:PD index = {[total PD (mmol/L)]/creatinine (mmol/L)} × BW^0.75^

Urine volume was estimated using creatinine concentration as a marker and assuming a daily creatinine excretion of 26 mg/kg of BW [37]. The estimated urinary creatinine excretion (0.9 mmol/kg of BW^0.75^) was included in the following equation to estimate the daily excretion of PD (mmol/kg of BW^0.75^):daily excretion of PD (dPD; mmol/kg of BW^0.75^) = PD index × 0.9.

From this, the amount of purines absorbed daily was estimated:daily absorbed purines (daP) = [dPD (mmol/kg of BW^0.75^) − 0.385 × BW^0.75^] + 0.85.

Microbial N (g of N/d) supply was estimated using the following equation:Microbial N (g of N/d) = (daP × 70)/(0.116 × 0.83 × 1000).

Nitrogen balance was calculated as follows: N intake = MN + UN + FN.

Total N intake (g/d) was determined by multiplying DMI and dietary N concentration. Milk N was calculated and reported as described by [38]: MN (g N/d) = Milk yield × (% CP in milk/6.38)/100.

Urinary N (UN) was estimated from the N concentration in urine and the daily urine volume (based on the urinary creatinine concentration, considering a daily creatinine excretion of 0.212 mmol/kg BW [39], while fecal N (FN) was estimated as the difference between N intake and N excreted in milk and urine.

### 2.6. Economic Analyses

The margin of milk income over feed costs was calculated for each cow and compared among dietary treatments. The payment scheme of the Prolesur dairy industry for Los Rios region, published in April 2020, was used to calculate the price per liter of milk produced by each cow according to its milk composition (Appendix A). The commercial costs of feed ingredients at the date of the trial (June 2020) were used for commercial concentrate, ground corn and soybean meal, whereas for green chopped perennial ryegrass, grass silage and sugar beet, the costs of production and transportation on farm were considered. In case of sugar beet, a total yield of 24 t DM/ha and a cost of US$2.672 were considered, which was the cost of production for sugar beets in Los Rios region for 2020.

### 2.7. Statistical Analyses

Prior statistical analyses, assumptions of normality and the homogeneity of variance of the data were checked. The data were analyzed using the mixed model procedure in SAS (SAS Institute Inc., Cary, NC, USA) to account for carryover effects according to the following model:y_ijklm_ = μ + S_i_ + A_(i)j_ + P_(i)k_ + T_l_ + C_m_ + e_(ijk)l_
where y_ijklm_ is an observation for each dependent variable; μ is the general mean; S_i_ is the fixed effect of the i^th^ treatment sequence (i = 1 to 6); A_(i)j_ is the random effect of the j^th^ cow in the i^th^ sequence; P_k_ is the fixed effect of the kth period (k = 1 to 3); T_l_ is the fixed effect of the lth treatment (l = 1 to 3); C_m_ is the fixed carryover effect from the previous period (C = 0, if period = 1); and e_(ijk)l_ is the random error. 

If carryover effects were not detected, a simplified model for a replicated Latin square was used: y_ijkl_ = μ + S_i_ + A_(i)j_ + P_(i)k_ + T_l_ + e_(ijk)l_
where y_ijkl_ is the observation for dependent variables; μ is the general mean; S_i_ is the random effect of the i^th^ square (i = 1 to 4); A_(i)j_ is the random effect of cow nested within square; P_(i)k_ is the fixed effect of the k^th^ period (k = 1 to 3); T_l_ is the fixed effect of the lth treatment (l = 1 to 3); and e_(ijk)l_ is the random error. Data for DM and nutrient intake, behavior, milk yield, milk composition, PD, MN and time of pH under a threshold were summarized by day. Data for VFA, NH_3_ and pH per hour were analyzed with the same model but including sampling time as a repeated measurement with cow as a subject, and the interaction between treatment and sampling time was also included. In the cases of VFA and NH_3_, the interaction between dietary treatment and time was not significant, and therefore it was removed from the model. The estimation method was REML, and the degrees of freedom method was Kenward-Roger. The variance–covariance structure that yielded the lowest corrected Akaike information criterion was compound symmetry, which was selected for the final model. Orthogonal polynomial contrasts were performed to determine the linear and quadratic effects of the inclusion of sugar beet. All data are reported as LSM ± SEM. Significance was declared at *p* ≤ 0.05 and trends at 0.05 < *p* ≤ 0.10.

## 3. Results

The effects of replacing ground corn with sugar beet root on DMI and nutrient intake of lactating dairy cows are reported in Table 2. The intakes of DM, NE_l_ and CP were linearly reduced with the inclusion of sugar beet (*p* < 0.05), whereas aNDFom was unaffected. The time spent eating was increased by 81 min/d and time under Other Activities (resting, social interactions, drinking, demonstration of estrus, and others) was reduced by 54 min/d for cows fed 100%-SB diet (*p* < 0.01), while rumination time was unaffected (*p* > 0.05). 

Milk production and composition were also affected by the replacement of ground corn with sugar beet (Table 3). Milk production was linearly reduced from 29.3 to 27.2 kg/d with the total replacement of corn with sugar beet (*p* = 0.003), as well as CP production (*p* = 0.022). In contrast, milk fat concentration was linearly increased from 4.25 to 4.51% for 100-C to 100-SB (*p* < 0.05), whereas milk CP concentration and fat production were not affected by dietary treatments, instead resulting in similar production when correcting milk for fat and protein content (FPCM). No differences were observed for BW and BCS changes in cows (*p* > 0.05), and final BW was linearly increased with SB inclusion (*p* = 0.004). Regarding the economic analysis, even though milk production of sugar beet (SB) supplemented cows was lower, the net income per cow was not affected by dietary treatments (*p* > 0.05), whereas feeding costs were linearly reduced from US$4.09 to US$2.99 with the replacement of ground corn with sugar beet roots, resulting in a linear increase in the margin over feed costs for SB supplemented cows.

Total VFA concentration in the rumen was unaffected by dietary treatments (*p* > 0.05). Replacing corn with sugar beet in the diet linearly reduced acetate molar proportion in the rumen (*p* = 0.047) and linearly increased butyrate molar proportion (*p* < 0.001). Rumen molar proportion of propionate was quadratically affected (*p* = 0.003), being greater for cows fed 50C-50SB than for those fed 100-C and 100-SB diets, and therefore, the acetate:propionate ratio tended to be quadratically affected (*p* = 0.064), with a reduction for cows fed the 50C-50SB diet (Table 4).

Urine purine derivates such as allantoin and uric acid concentrations, as well as estimated microbial N, were unaffected by dietary treatments, whereas creatinine concentration was linearly increased by 3 mmol/d (*p* = 0.018) with 100-C to 100-SB diets. Milk N, MUN, rumen NH_3_ and urinary N were linearly reduced when replacing corn with sugar beet by 9 g/d, 1.6 mg/dL, 4.9 mmol/L and 11 g/d, respectively (*p* < 0.05), whereas fecal N and NUE was quadratically affected (*p* < 0.1). In case of fecal N, it was reduced by 15.5 g/d with a partial replacement, and then, it increased 5.2 g/d with the total replacement of corn with sugar beet. Conversely, NUE increased by 0.8 percent units and then reduced 1.0 percent unit with partial and total replacement, respectively.

Mean rumen pH was quadratically affected by dietary treatments, being reduced by 0.1 for cows supplemented with 50C-50SB and then increasing with the 100-SB diet. Nevertheless, times of pH under 5.8, between 5.8 and 6.2 and above 6.2 were unaffected by dietary treatments (Table 5 and Figure 1).

## 4. Discussion

Cereal grains have increased in cost, and therefore dairy farmers try to find alternatives to provide energy in the rumen. Sugar beet roots have a high energy content and may be a sound alternative for partially or totally replacing ground corn in the diet of dairy cows. However, feeding SB roots to dairy cattle to replace a portion of the grain in the ration is a concept that has not received sufficient attention [17]. Thus, the aim of this study was to determine DMI, feeding behavior, rumen fermentation, milk production responses and N partitioning when replacing ground corn with increasing levels of sugar beet in pasture-fed lactating dairy cow diets.

### 4.1. Intake, Performance and Milk Composition

The reduction in voluntary DMI may be attributed to the high water content of SB, which results in physical limitations for ingestion because of the bulkiness and high moisture content of sugar beet roots that increase the ‘as-fed’ amount that cows have to ingest [17]. Eriksson et al. [27] observed a 1 kg DM reduction in alfalfa/grass silage intake when lactating dairy cows were offered fodder beets instead of barley. Castillo-Umaña et al. [40] observed a reduction in total DMI when offering another root crop such as summer turnips. The physical structure of root crops (hard and difficult to eat [41]) may also affect DMI and ingestive behavior, increasing the time spent eating. In this study, eating time was increased by 81 min/d when comparing cows supplemented with sugar beets instead of ground corn. Previous studies [12,40] observed that cows supplemented with a root crop (summer turnips or swedes) spent more time eating compared with cows offered other forages similar in water content to root crops and another diet composed of silage and concentrate. Thus, we suggest that the reduction in DMI of sugar beet-fed cows is a result of the combined effect of the low water content of the diet and the physical structure of roots. 

The reduction in milk production agrees with the lower DMI and NE_l_ intake of sugar beet-supplemented cows. A reduction in milk yield has been previously reported when replacing barley with fodder beets [27], which was also associated with a reduction in DMI for beet-supplemented cows. In contrast, no reduction in milk yield and DMI when replacing corn and barley by sugar beets has been previously reported. The lack of differences in milk protein content is in line with the similar estimated microbial nitrogen, one of the main sources of amino acid duodenal protein supply in dairy cows [42]. The linear increase in the milk fat content of sugar beet supplemented cows is explained by the greater rumen butyrate concentration, as butyrate is one of the main precursors for the synthesis of milk fatty acids in the mammary gland [43]. The increase in milk fat content explained the lack of differences in FPCM. The payment scheme (Appendix A) used for the calculation of milk price considered a bonus for the concentration of milk solids, and as milk fat and crude protein contents were statistically and numerically greater for SB supplemented cows, the price obtained for SB milk was greater, and the net income per cow was not different. Thus, considering the greater feed cost for corn-supplemented cows compared with a partial or total replacement, the margin over feed cost was increased with the inclusion of sugar beets in the diet. The potential cost savings of including sugar beets in dairy feeding programs has been previously suggested [7,17]; however, both studies recognized the need for further research. This study helps to overcome this lack of information. It is important to note that the incorporation of SB in dairy systems from humid-temperate regions may not imply competition with the sugar industry; instead, farmers could produce their own SB on farm, as farm-produced crops help reduce feeding costs [44]. 

### 4.2. Rumen Metabolism and N Excretions

It is worth mentioning that total VFA concentrations are lower than values usually reported in the literature and may be explained as the different sampling procedure used in this study (stomach tubes are inserted into the rumen) compared with the most common procedure (rumen cannulation), and the reported data represent VFA values in the reticulum. Therefore, comparisons of rumen metabolite concentrations should be carefully noted as reticular salivary presence is highly variable, and although VFA relative percentages were not affected, total VFA concentrations may be. The lack of change in total VFA concentration was the same when barley was replaced by fodder beets [27] and sucrose from sugar beet molasses replaced starch from wheat [45]. In the current study, all diets were formulated to be isoenergetic and had similar digestibility. This may explain the lack of differences in rumen VFA concentrations, as total VFA concentration in the rumen is associated with the digestibility of the diet [46]. The inclusion of the sugar beets increased the relative percentage of butyrate at the expense of acetate, which is in accordance with when sugar-rich diets are offered to dairy cows [7,47] and reported when including fodder beet [27,48] or swedes [11,12]. This increase in rumen butyrate has resulted in increased blood betahydroxbutyrate (BHB) [49], although other studies reported no effect of increased BHB [12]. In the current study, there were no differences in BW and BCS change; in addition, cows were not under negative energy balance, so elevated BHB was a consequence of greater rumen butyrate production. The effects of increased BHB due to increased rumen butyrate on overall productivity and health are still unknown and require further investigation [7,49].

It has been thought that beets may not be suitable as a feed ingredient due to the high sugar content compared with starch. Sugar has been associated with a reduction in rumen pH, fiber digestion and microbial yield and therefore has been rejected from feeding programs [17]. In the current study, lower mean rumen pH was observed in cows fed corn and sugar beet, but none of the dietary treatments presented risk of subacute rumen acidosis, that is rumen pH below 5.8 [50]. Although sugars are readily and extensively broken down in the rumen, they do not mimic the pH-lowering effect of starch and do not produce lactate and rumen pH reduction as starch does [47]. Furthermore, it has been proposed that the bulkiness of root crops may increase saliva production and therefore buffer changes in rumen pH [12]. Thus, the results presented in this study agree with those of Oba [7] and Evans and Messerschmidt [17] that lactating dairy cows can be supplemented with high concentrations of sugar (21% on DM basis for 100SB) without negatively affecting rumen pH. Regarding N metabolism, although diets were formulated to be iso-nitrogenous, N intake was linearly reduced with the replacement of corn with sugar beet (28.8 g/d less for SB supplemented cows), mainly due to the lower DMI (−1.1 kg DM). This lower N intake did not affect estimated MN synthesis, increasing NUE for cows fed the 50C-50SB diet. A previous study showed that fresh fodder beets supported microbial growth to the same extent as fresh potatoes (rich in starch) [26]. High microbial crude protein has been observed in fodder beet diets due to the higher rumen fluid passage rate. Since sugar beet also has a low dry matter content, it may has increased the fluid passage rate and therefore decreased the MN turnover [51]. Despite the differences in N intake between treatments, microbial N (g N/d) was not affected by treatment, reflecting that the greater N intake was not utilized for the synthesis of microbial N. In fact, energy supply is the main factor limiting microbial growth in the rumen of cattle grazing temperate pastures [52,53]. Instead, the lower N intake resulted in lower rumen NH_3_ concentration, being positively correlated with MUN. Milk urea N and NH_3_ in the rumen have been associated with the lower efficiency of microbial protein synthesis [54]. The surplus of N intake compared with requirements is usually excreted into the environment [55]. The greater urinary N excretion is in part a consequence of greater concentrations of NH_3_ in the rumen. Once NH_3_ is produced by ruminal bacteria, it can be used to build microbial protein (energy-dependent process) or be transported to the liver (low energy availability in the rumen) to be converted into urea; then, it may be excreted through the urine [55,56,57] or recycled into the rumen along with saliva [42]. Nitrogen excretion through the urine and dung are important in terms of environmental pollution because they are an important N source for N_2_O emissions in pasture-based livestock systems, a powerful greenhouse gas whose global warming potential (GWP) is greater than CO_2_ and CH_4_ [58]. Urinary N is mainly excreted as NH_3_, whose oxidation and linked nitrifier denitrification are the major processes generating N_2_O and therefore contribute to indirect emissions of N_2_O [38]. As potential N_2_O emissions from urine are five times greater than dung N, and therefore, nutritional strategies should be focused on shifting the N excretion from urine to dung or reducing urinary N excretion, with the aim to reduce N_2_O [59]. Field studies indicate that 3% to 15% of total excretal N is lost via NH_3_ volatilization, whereas the fraction of urinary N released as N_2_O varies from <1% to more than 10% [60]. Therefore, the greater urinary N excretion for corn-supplemented cows may result in greater N_2_O emissions compared with SB supplemented cows. Furthermore, Dijskstra et al. [60] stated that increasing urine volume appears a promising N_2_O mitigation strategy, particularly in pasture-based systems. Although the data are not shown, estimated urine volume (based on creatinine concentrations) was greater for SB-fed cows, and therefore it may help reduce N_2_O emissions. 

## 5. Conclusions

The replacement of ground corn with sugar beet as a supplement in the diet of pasture-fed lactating dairy cows reduced DM intake and milk production. Nevertheless, milk fat concentration was increased, and therefore, when correcting milk yield for the fat and protein content, no difference was observed. Furthermore, feeding cost was considerably reduced, and the price per kg of milk was increased, thereby increasing the margin over feed costs. No negative effects on total volatile fatty acids and rumen pH were observed, and the greater rumen butyrate supports the increase in milk fat content. Furthermore, SB-supplemented cows reduced urinary N excretions, and thus, this supplementation may contribute to reducing N_2_O emissions from dairy systems. Thus, it can be concluded that using sugar beet roots as an energy supplement can be a suitable alternative to ground corn in pasture-fed lactating dairy cows.

## Figures and Tables

**Figure 1 animals-12-01927-f001:**
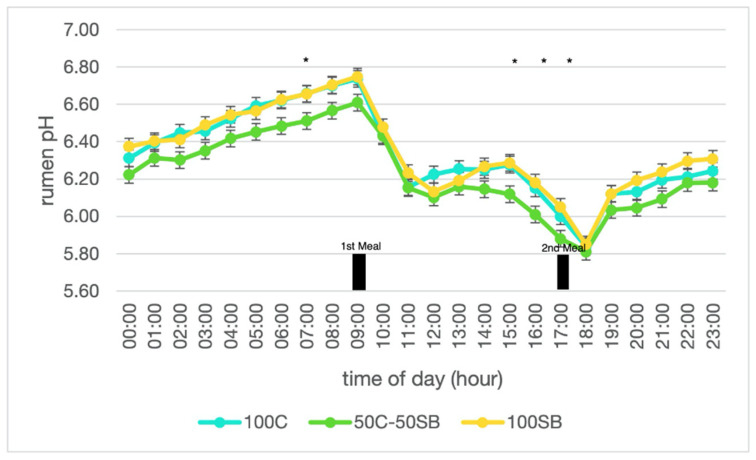
The effects of replacing ground corn with sugar beet root on the daily variations in the rumen pH of early-lactating dairy cows. * Significant differences (*p* < 0.05) among dietary treatments. Error bars denote standard error of the mean; 100C, control diet with ground corn as energy supplement; 50C-SB, diet with 50% ground corn and 50% sugar beets as energy supplements; 100SB, diet with 100% sugar beet as energy supplement.

**Table 1 animals-12-01927-t001:** The nutrient concentrations of the feed ingredients and diets (% on DM basis if not otherwise stated).

	DM ^1^	Ash	CP	EE	aNDFom	ADFom	WSC	Starch	NE_l_	Ca	P
Ingredients											
Concentrate	89.3	8.97	15.5	4.06	32.16	14.40	4.43	34.3	1.77	1.37	0.64
Soybean meal	88.0	7.28	50.3	0.98	9.1	6.8	12.3	2.7	1.89	0.36	0.68
Ground Corn	86.1	1.42	7.9	3.80	11.5	2.8	1.63	71.3	1.84	0.04	0.32
Sugar Beet	22.1	1.67	4.9	0.38	12.5	5.7	69.5	3.11	1.75	0.33	0.07
Perennial Ryegrass	15.6	10.08	23.5	3.8	46.9	25.8	9.30	2.23	1.65	0.49	0.35
Grass Silage	43.2	8.38	14.1	2.9	48.5	30.6	3.32	2.67	1.62	0.66	0.32
Diets											
100C	51.3	8.5	17.3	3.3	36.1	20.2	5.2	20.2	1.68	0.73	0.40
50C-50SB	44.2	8.4	16.8	3.0	35.9	20.3	13.2	12.8	1.68	0.76	0.37
100SB	37.3	8.4	16.4	2.6	35.8	20.5	20.9	5.5	1.67	0.78	0.34

^1^ DM, dry matter; CP, crude protein; EE, ether extract; aNDFom, neutral detergent fiber treated with a heat stable amylase; ADFom, acid detergent fiber, WSC, water soluble carbohydrates; NE_l_, net energy of lactation; ca, calcium; P, phosphorus; 100C, control diet with ground corn as energy supplement; 50C-SB, diet with 50% ground corn and 50% sugar beets as energy supplements; 100SB, diet with 100% sugar beet as energy supplement.

**Table 2 animals-12-01927-t002:** The effects of replacing ground corn with sugar beet root on nutrient intake and behavior of early- lactating dairy cows.

					*p*-Value	
100-C		50C-50SB	100-SB	SEM	Linear	Quad
Intake						
DM (kg DM/d)	22.4	21.4	21.3	0.43	0.031	0.289
NE_l_ (Mcal/d)	37.7	35.7	35.5	0.64	0.005	0.095
CP (kg DM/d)	3.8	3.6	3.6	0.07	0.013	0.173
aNDFom (kg DM/d)	7.6	7.6	7.7	0.13	0.5	0.462
Eating (min/d)	392	403	473	12.1	<0.001	0.009
Ruminating (min/d)	456	452	428	16.7	0.148	0.557
Other activities (min/d)	330	323	276	16.4	0.009	0.233

DM, dry matter; CP, crude protein; NE_l_, net energy of lactation; aNDFom, neutral detergent fiber with a heat stable amylase; SEM, standard error of the mean; 100C, control diet with ground corn as energy supplement; 50C-SB, diet with 50% ground corn and 50% sugar beets as energy supplements; 100SB, diet with 100% sugar beet as energy supplement.

**Table 3 animals-12-01927-t003:** The effects of replacing ground corn with sugar beet root on milk production and composition of early-lactating dairy cows.

					*p*-Value	
	100-C	50C-50SB	100-SB	SEM	Linear	Quad
Milk yield (kg/d)	29.3	28.5	27.2	1.17	0.003	0.629
MY/DMI (kg)	1.31	1.33	1.28	0.069	0.322	0.255
Fat (%)	4.25	4.45	4.51	0.129	0.045	0.519
Fat (kg/d)	1.22	1.25	1.21	0.047	0.694	0.301
CP (%)	3.47	3.53	3.56	0.112	0.113	0.668
CP (kg/d)	1.01	1.00	0.96	0.018	0.022	0.412
FPCM (kg/d)	30.2	30.3	29.1	0.68	0.104	0.311
FPCM/DMI (kg)	1.36	1.41	1.37	0.041	0.849	0.224
BCS initial	2.64	2.56	2.64	0.062	0.994	0.178
BCS final	2.69	2.68	2.63	0.074	0.498	0.840
BCS change	0.05	0.12	−0.01	0.095	0.645	0.367
BW initial (kg)	598	604	607	15.3	0.539	0.937
BW final (kg)	607	615	622	15.5	0.004	0.872
BW change (kg)	9	11	15	0.1	0.983	0.692
Net Income (US$/cow/d)	10.4	10.4	10.0	0.22	0.108	0.252
Feed Cost (US$/cow/d)	4.1	3.5	3.0	0.07	<0.001	0.182
MOFC (US$/cow/d)	6.2	6.9	7.0	0.18	0.004	0.115

MY/DMI, milk yield to dry matter intake ratio; CP, crude protein; FCPM, Fat and protein corrected milk; FPCM/DMI, fat and protein corrected milk to dry matter ratio; BCS, body condition score; BW, body weight; MOFC, margin over feed costs; SEM standard error of the mean; 100C, control diet with ground corn as energy supplement; 50C-SB, diet with 50% ground corn and 50% sugar beets as energy supplements; 100SB, diet with 100% sugar beet as energy supplement.

**Table 4 animals-12-01927-t004:** The effects of replacing ground corn with sugar beet root on rumen volatile fatty acids (VFA), purine derivatives (PD) and estimated microbial nitrogen (N) of early-lactating dairy cows.

					*p*-Value	
	100-C	50C-50SB	100-SB	SEM	Linear	Quad
Total VFA (mM)	64.3	67.9	65.2	2.21	0.748	0.229
VFA (mol/100 mol)						
Acetate	55.7	54.4	54.0	0.62	0.047	0.548
Butyrate	16.9	17.3	19.3	0.47	<0.001	0.046
Propionate	24.1	25.3	23.7	0.41	0.357	0.003
Isobutyrate	1.1	1.2	1.1	0.04	0.713	0.372
Isovalerate	1.0	1.1	1.1	0.05	0.560	0.386
Valerate	0.8	0.9	1.1	0.10	0.103	0.827
Minor VFA	2.9	3.2	3.3	0.17	0.211	0.619
Acetate:Propionate	2.33	2.18	2.3	0.06	0.632	0.064
Ammonia (mmol/L)	18.7	16.4	13.8	0.77	<0.001	0.891
Allantoin (mmol/d)	335	364	365	15.1	0.218	0.458
Uric acid (mmol/d)	66	62	76	9.7	0.372	0.327
Creatinine (mmol/d)	139	140	142	1.2	0.018	0.319
PD excretion (mmol/d)	401	427	440	22.1	0.106	0.718
Microbial N (g/d)	250.8	270.4	277.3	10.36	0.123	0.625
Milk N (g/d)	159.3	156.0	150.6	2.78	0.009	0.707
Urinary N (g/d)	271.4	259.1	260.1	4.46	0.026	0.148
Milk urea N (mg/dL)	11.2	10.4	9.6	0.46	<0.001	0.918
Fecal N (g/d)	174.7	159.2	164.4	6.56	0.112	0.069
NUE (%)	26.4	27.2	26.2	0.66	0.792	0.096

NUE, Nitrogen use efficiency; SEM, standard error of the mean; 100C, control diet with ground corn as energy supplement; 50C-SB, diet with 50% ground corn and 50% sugar beets as energy supplements; 100SB, diet with 100% sugar beet as energy supplement.

**Table 5 animals-12-01927-t005:** The effects of replacing ground corn with sugar beet root on the rumen pH of early-lactating dairy cows.

				SEM	*p*-Value	
	100-C	50SB-50SB	100-SB		Linear	Quad
pH	6.36	6.26	6.37	0.018	0.190	<0.001
pH < 5.8 (min/d)	22	25	7	11.6	0.413	0.682
pH 5.8–6.2 (min/d)	405	599	437	91.4	0.687	0.234
pH > 6.2 (min/d)	1006	841	994	92.4	0.859	0.242

SEM, standard error of the mean; 100C, control diet with ground corn as energy supplement; 50C-SB, diet with 50% ground corn and 50% sugar beets as energy supplements; 100SB, diet with 100% sugar beet as energy supplement.

## Data Availability

The data presented in this study are available on request from the corresponding author.

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
