# Peer review of "The Replacement of Ground Corn with Sugar Beet in the Diet of Pasture-Fed Lactating Dairy Cows and Its Effect on Productive Performance and Rumen Metabolism"

_animals, 2022, doi:10.3390/ani12151927_

Round 1
Reviewer 1 Report
Review report for “Replacement of ground corn by sugar beet in the diet of pasture-fed lactating dairy cows and its effect on productive performance and rumen metabolism.”
Dear authors,
This is a well written paper with interesting findings and a broad range of methodologies used. I have only a few comments to be addressed:
1. The discussion goes straight into the findings. It would be beneficial to include a short paragraph at the start to remind your readers of the big picture and aim of the study.
2. I would like to see brief mention of the rumen microbiome in terms of composition/function and how this might be affected by high sugar diets compared with starch. There is a paper here which may help: https://www.frontiersin.org/articles/10.3389/fnut.2021.727714/full. Sugar beet is also rich in fibre and bioactive components which may play a role.
3. Finally, I think the paper would benefit from some discussion on the feasibility of using sugar beet root/whole crop on a commercial scale. Sugar beet pulp is used as a by-product of the sugar industry, but do you see any issues with competition for substrate if you are proposing the use of the entire sugar beet crop as a product for ruminant feed? You mention that cereal crops have high demand from other industries so there is need for alternative but it is worth mentioning there is also demand on sugar beet crops.
Best wishes.
Author Response
This is a well written paper with interesting findings and a broad range of methodologies used. I have only a few comments to be addressed:
Many thanks for your comment and support to the work we've done
- The discussion goes straight into the findings. It would be beneficial to include a short paragraph at the start to remind your readers of the big picture and aim of the study.
Thanks, we added a short paragraph regarding it (please se lines 505-512)
- I would like to see brief mention of the rumen microbiome in terms of composition/function and how this might be affected by high sugar diets compared with starch. There is a paper here which may help: https://www.frontiersin.org/articles/10.3389/fnut.2021.727714/full. Sugar beet is also rich in fibre and bioactive components which may play a role
Thanks we added a statment about how micribiome is changed in sugar rich diets, see lines 70-73
- Finally, I think the paper would benefit from some discussion on the feasibility of using sugar beet root/whole crop on a commercial scale. Sugar beet pulp is used as a by-product of the sugar industry, but do you see any issues with competition for substrate if you are proposing the use of the entire sugar beet crop as a product for ruminant feed? You mention that cereal crops have high demand from other industries so there is need for alternative but it is worth mentioning there is also demand on sugar beet crops.
Included, please see lines 553-556.
Reviewer 2 Report
This is generally a good study which have merit and should be puplished. However, the manuscript was written sloppily and needs deep revision.
Introduction: L49-53, competition with food production? L76-92, maybe you could add a short overview about the different pathways of starch and sugar degradation and the possible risks associated with them.
Materials and methods: L104-106, so, in each experimental group, cows covering the whole range of milk production were included? Please state this clearer? L123, delete "b". Tab. 1, consistently use aNDFom and ADFom abbreviations as you stated in 2.2 that this was the way NDF and ADF have been analysed; Tab. 1 (footnotes), "treated" with a heat stable amylase. L136-138, needs to be revised. L187, words are incomplete. L196, "propionate" instead of "proportionate". L230, please give a reference for the used factor. Statistical analysis, please explain P(i)k, why not Pk? Delete "m" from the second model (Yijkl). L266-267, explain "the interaction of treatment and repeated measurement". Did you check residuals for Gaussian distribution?
Results: L279, NEL decreased significantly, so please revise your statement. L281, sentence is incomplete. L296, words are missing. L323, "FN", please use abbreviations consequently or consequently not. Fig. 1, y- and x-axis titles are missing.
Discussion: Headline "4. Discussion" is missing. L359-361, this is exactly what you wrote before - not necessary to repeat.
Author Response
This is generally a good study which have merit and should be puplished. However, the manuscript was written sloppily and needs deep revision.
A: thanks for you opinion and comments. We've improved spelling and grammar of the manuscript and considered all your suggestions:
Introduction:
L49-53, competition with food production?
Amended
L76-92, maybe you could add a short overview about the different pathways of starch and sugar degradation and the possible risks associated with them.
Included, lines 62-69
Materials and methods:
L104-106, so, in each experimental group, cows covering the whole range of milk production were included? Please state this clearer?
Clarified, lines 143-144
L123, delete "b".
Thanks, deleted
Tab. 1, consistently use aNDFom and ADFom abbreviations as you stated in 2.2 that this was the way NDF and ADF have been analysed;
Amended
Tab. 1 (footnotes), "treated" with a heat stable amylase.
Amended
L136-138, needs to be revised.
Amended.
L187, words are incomplete.
Amended
L196, "propionate" instead of "proportionate".
Amended
L230, please give a reference for the used factor.
Included
Statistical analysis, please explain P(i)k, why not Pk? Delete "m" from the second model (Yijkl).
Thanks for the observations, amended
L266-267, explain "the interaction of treatment and repeated measurement".
Amended
Did you check residuals for Gaussian distribution?
A: yes, included
Results:
L279, NEL decreased significantly, so please revise your statement.
Thanks, amended
L281, sentence is incomplete.
amended
L296, words are missing.
amended
L323, "FN", please use abbreviations consequently or consequently not.
Amended
Fig. 1, y- and x-axis titles are missing.
Amended
Discussion:
Headline "4. Discussion" is missing.
Included
L359-361, this is exactly what you wrote before - not necessary to repeat.
Sentence deleted
Reviewer 3 Report
Line 14: …sound alternatives “to” partially…
Line 16: …ground corn “with” fresh sugar beet..
Line 18: …with sugar beet roots “had” reduced dry matter intake and milk production “compared to control cows”, fat…
Line 21: …increased “for sugar beets”.
Line 22: Do you mean N2O excretion? Because N by itself is not a greenhouse gas.
Line 33-37: p-values needed
Line 74: I “t” a metric ton or imperial ton?
Line 101-102: Diet fo “15 d uniformity period” should be defined.
Line 106: suggest …replicated “(n=4)” 3x3 latin square… like it is in the abstract
Line 115: Define what were the grasses in the grass pasture? State how was pasture intake known/measured? State how animals were housed on pasture. State whether animals were individually fed or group fed.
Line 137: incomplete sentence
Line 186: How was saliva contamination handled? Was it removed?
Line 187: d 20mental?
Line 195: More detail needed for VFA analysis. Extraction procedure? Type of GC, column type and length, temperature program
Line 196: propionate
Line 203: Have pH measurements over long periods of time with only 1 calibration been validated with this equipment?
Line 211: more detail needed for HPLC analysis.
Line 230: milk protein measurement/calculation is not clear. Do you mean that Milk N was calculated from the measured milk protein (using NIR) using a conversion factor of protein to N of 6.38? What is the reference for 6.38?
Line 253: more detail needed about how treatment sequences were allotted to animals (in line 100-107?)
Line 261: more detail needed about how cows were allotted to squares (in line 100-107?)
Line 279: NEl was linearly decreased (P = 0.005). Re-word sentence
Line 281: what are “other activities”
Line 294: CP production was decreased linearly (P = 0.022). Re-word sentence
Line 296: Final BW was linearly increased by SB (P = 0.004). Re-word sentence.
Figure 1: axis labels needed. what is the time on the x axis relative to? it seems as if it is time (ie starts at 1 AM). should it be relative to feeding (ie, time 0 is feeding time)?
Line 368: what in particular about the physical structure of beets that would restrict intake? Any possible metabolite or chemical that could be causing it?
Line 374-376: more detail needed. What in particular about the microbial N supply is causing no change in milk protein?
Line 395: stomach tubes are inserted into the rumen
Line 405-406: sentence repeated
Line 408: final BW was linearly increased by sugar beet (P = 0.004). Re-word sentence
Line 417: any speculation why the combination of corn and sugar beet would have the lowest rumen pH compared to corn alone or sugar beet alone?
Line 434: how would increased passage rate increase MN turnover?
Line 449: incomplete sentence. ..whose GWP is….?
Line 449-450: N content of urine may be 5x greater than feces, but the form of N in urine is not predominantly N2O. Urinary N is mostly NH3, which is not a greenhouse gas. Form of fecal N is mostly N2O.
Author Response
Thanks fort all your comments and thorough review that certainly helped improving the manuscript. Please see our replies to your comments
Line 14: …sound alternatives “to” partially…
included
Line 16: …ground corn “with” fresh sugar beet..
Amended
Line 18: …with sugar beet roots “had” reduced dry matter intake and milk production “compared to control cows”, fat…
Amended
Line 21: …increased “for sugar beets”.
Amended
Line 22: Do you mean N2O excretion? Because N by itself is not a greenhouse gas.
Amended
Line 33-37: p-values needed
Included
Line 74: I “t” a metric ton or imperial ton?
Changed to kg
Line 101-102: Diet fo “15 d uniformity period” should be defined.
amended
Line 106: suggest …replicated “(n=4)” 3x3 latin square… like it is in the abstract
Added
Line 115: Define what were the grasses in the grass pasture? State how was pasture intake known/measured? State how animals were housed on pasture. State whether animals were individually fed or group fed.
A: Explained in lines 152-153 and 185-190
Line 137: incomplete sentence
amended
Line 186: How was saliva contamination handled? Was it removed?
A: In order to reduce saliva contamination, the first portion of the liquor collected was discarded (lines 255-256).
line 187: d 20mental?
amended
Line 195: More detail needed for VFA analysis. Extraction procedure? Type of GC, column type and length, temperature program
Included (262-264)
Line 196: propionate
amended
Line 203: Have pH measurements over long periods of time with only 1 calibration been validated with this equipment?
This equipment is certificated by the sensor manufacturers and in pre-trials we validated them in fistulated cows.
Line 211: more detail needed for HPLC analysis.
Inluded (lines 279-289)
Line 230: milk protein measurement/calculation is not clear. Do you mean that Milk N was calculated from the measured milk protein (using NIR) using a conversion factor of protein to N of 6.38? What is the reference for 6.38?
Amended (l 404)
Line 253: more detail needed about how treatment sequences were allotted to animals (in line 100-107?)
Included, lines 147 - 150
Line 261: more detail needed about how cows were allotted to squares (in line 100-107?)
Included, lines 146-147
Line 279: NEl was linearly decreased (P = 0.005). Re-word sentence
Amended
Line 281: what are “other activities”
Stated in lines 212-213
Line 294: CP production was decreased linearly (P = 0.022). Re-word sentence
Amended
Line 296: Final BW was linearly increased by SB (P = 0.004). Re-word sentence.
Amended
Figure 1: axis labels needed. what is the time on the x axis relative to? it seems as if it is time (ie starts at 1 AM). should it be relative to feeding (ie, time 0 is feeding time)?
Axis labels were included
Line 368: what in particular about the physical structure of beets that would restrict intake? Any possible metabolite or chemical that could be causing it?
A: root crops are hard and therefore difficult to eat, included in the text (line 575). We don’t believe that in case of SB roots it is associated with metabolites or chemical compounds
Line 374-376: more detail needed. What in particular about the microbial N supply is causing no change in milk protein?
A: It caused no effect as microbial N was similar among dietary treatments (lines 656-660)
Line 395: stomach tubes are inserted into the rumen
Amended
Line 405-406: sentence repeated
Deleted
Line 408: final BW was linearly increased by sugar beet (P = 0.004). Re-word sentence
Although final BW was linearly increased, BW change was unaffected
Line 417: any speculation why the combination of corn and sugar beet would have the lowest rumen pH compared to corn alone or sugar beet alone?
Even though pH values are statistically lower it occurred only at four times of the day with a difference of 0.1 units, being not biologically relevant
Line 434: how would increased passage rate increase MN turnover?
Thanks for the observation, it was ‘decreased’
Line 449: incomplete sentence. ..whose GWP is….?
Sentence completed
Line 449-450: N content of urine may be 5x greater than feces, but the form of N in urine is not predominantly N2O. Urinary N is mostly NH3, which is not a greenhouse gas. Form of fecal N is mostly N2O.
A: this is true, however ammonia oxidation and linked nitrifier denitrification are the major processes generating N2O. Sentence included to clarify the concept (679-682)
Round 2
Reviewer 3 Report
Line 13, 396: …have increased “in” cost…
Line 61-75: Too much detail for an introduction. Suggest removing pathway information and bacterial changes.
Line 122: …[38] “and balanced” for residual…
Line 132: The use of the word “pasture” implies that cattle grazed the ryegrass. Line 159 says it was harvested. Was it fed fresh daily? Was it green chop?
Line 265: Not clear. How milk protein and N were determined needs to be better described.
Line 321: “Other Activities” need to be defined.
Figure 1: What is time in figure 1? Time relative to when animals were fed needs to be defined in the graph.
Line 509-511: the amount of NH3 from urine that is volatilized versus how much is nitrified into N2O needs to be stated and reference included.
Author Response
Dear reviewer,
Thanks again for your comments, here are our answers:
Line 13, 396: …have increased “in” cost…
A: amended
Line 61-75: Too much detail for an introduction. Suggest removing pathway information and bacterial changes.
A: We agree, however another reviewer and the section editor requested us to include fermentation pathways and effects on rumen microbiome
Line 122: …[38] “and balanced” for residual…
A: amended
Line 132: The use of the word “pasture” implies that cattle grazed the ryegrass. Line 159 says it was harvested. Was it fed fresh daily? Was it green chop?
A: ameded throughout the text
Line 265: Not clear. How milk protein and N were determined needs to be better described.
A: amended, equation included
Line 321: “Other Activities” need to be defined.
A: defined
Figure 1: What is time in figure 1? Time relative to when animals were fed needs to be defined in the graph.
A: amended. X axis is time of the day, we included feeding times
Line 509-511: the amount of NH3 from urine that is volatilized versus how much is nitrified into N2O needs to be stated and reference included.
A: It has been stated with a reference, please see lines 512-514